# Predicting Lesion Growth and Patient Survival in Colorectal Cancer Patients using Deep Neural Networks

**Alexander Katzmann, Alexander Mühlberg, Michael Sühling**
*Siemens Healthcare GmbH
Department CT R&D Image Analytics
91301 Forchheim, Germany

**Dominik Nörenberg[a], Julian Walter Holch[b], Volker Heinemann[b]**
University Hospital Großhadern, Ludwig-Maximilians-University München
[a]Department of Radiology
[b]Department of Internal Medicine III, Comprehensive Cancer Center
Marchioninistrasse 15, 81377 Munich, Germany

**Horst-Michael Groß**
Neuroinformatics and Cognitive Robotics Lab
University of Technology Ilmenau
98693 Ilmenau, Germany

alexander.katzmann@siemens-healthineers.com

## Abstract

Being responsible for over 50,000 death per year within the U.S. alone, colorectal cancer (CRC) is the second leading cause of cancer related deaths in industry nations with increasing prevalence. Within the scope of personalized medicine, precise estimates on future progress are crucial. We thus propose a novel deep learning based system using deep convolutional sparse autoencoders for estimating future lesion growth for CRC liver lesions based on single slice CT tumor images for early therapy assessment. Furthermore, we show that our system can be used for one-year survival prediction in CRC patients. While state of the art treatment assessment (RECIST) is premised on retrospective lesion analysis, our proposed system delivers an estimate on future response, thus prospectively allowing to adapt therapy before further progress. We compare our system to single-lesion assessment through RECIST diameter and Radiomics. With our approach we archieve a $\phi$-coefficient of 40.0 % compared to 27.3 % / 29.4 % and an AUC of .784 vs .744/.737 for growth prediction, as well as a $\phi$-coefficient of 44.9 % vs 32.1 % / 18.0 % and an AUC of .710 vs. .688/.568 for survival prediction.

## 1 Introduction

In industry nations, cancer is the second leading cause of death following cardiovascular diseases, being responsible for approximately 21 % of all deaths. With increasing rates in industrial nations,

---

*The concepts and information presented in this article are based on research and are not commercially available.

1st Conference on Medical Imaging with Deep Learning (MIDL 2018), Amsterdam, The Netherlands.

CRC is responsible for over 50,000 deaths in 2017 within the US alone, making it a leading cause of cancer-related deaths worldwide (Ame (2017)). When diagnosed in an early stage, colorectal cancer can be treated well, whereas curing chances for the metastatic disease decrease rapidly. While overall mortality has decreased in the last years, 5-year survival for patients with metastatic CRC is still poor with approximately 12 % (Mody & Bekaii-Saab (2018)). At the initial diagnosis distant metastases are present in approximately 25 % of the patients, and another half of the patients develop metastases within the treatment Vatandoust *et al.* (2015). Thus, treating colorectal cancer commonly involves systematic treatment of liver metastases by chemotherapy, surgical intervention and continous observation.

For solid tumors, the Response Evaluation Criteria in Solid Tumors (RECIST) provide „a standardized set of rules for response assessment using tumor shrinkage" (Eisenhauer *et al.* (2009); Schwartz *et al.* (2016)). However, RECIST requires a large increase in lesions, as well as a long delay to detect disease progression. Also tumor growth may not neccessarily be linked to disease progression, as e.g. immuno-oncologic therapies are linked to a pattern called pseudoprogression, marked by temporary lesion growth under treatment response Chiou & Burotto (2015). Oxnard *et al.* (2012) conclude that current criteria for progression may not adequately capture disease biology. Thus, having a high precision early lesion estimate on future growth would be of high clinical value, allowing to prematurely double-check, and thus potentially even prepone treatment decisions.

We demonstrate that sparse lesion characterizations generated through deep convolutional dimensionality reduction are significantly predictive for future lesion's and patient's outcome. We therefore propose an early response assessment based on deep convolutional neural networks, showing significant correlation with outcome parameters, as well as predictive power for future lesion therapy response. Therefore...:

- We present a novel approach for CT liver lesion assessment, capable of predicting tumor growth with significant superiority to assessment through RECIST diameter and hand-crafted Radiomics features
- We show, that the same approach can be used for patient survival prediction, again outperforming prediction through RECIST diameter and Radiomics features
- We present a reasoning mechanism based on saliency maps, allowing to determine tumor growth patterns and possibly allow to gain new insights on tumor growth patterns

## 2 Data

The used data stems from two sources, one clinical and one radiologic data set. The radiologic data (Dataset A) is available for 116 patients, while clinical data could only be provided for 78 of these (Dataset B). For the presented algorithmic approaches we utilized the radiologic data exclusively, as well as a combination of both datasets (Dataset C):

- **Dataset A** - the radiologic base dataset. It consists of 1235 computed tomography images in DICOM format as well as high-quality, fully-volumetric segmentations of liver lesions. An example lesion segmentation can be found in Fig. 1. The dataset further contains radiologically extractable information, e.g. longest tumor diameter, longest diameter in one slice (referred to as *RECIST diameter*) and tumor volume. Since RECIST based labels get extracted from the radiologic ground truth (see below), at least two consecutive timepoints per structure get merged into one sample (Dataset A1) for prediction. Since we expect the extrapolation of previous growth as likely predictive for future progress, using two consecutive timepoints for prediction plus another for label extraction we receive another set which is used for our experiments on tumor growth estimation (Dataset A2).
- **Dataset B** - the clinical base dataset. It is acquired from a variety of sources and thus contains demographic data, blood values, tumor markers (e.g. Ca19.9), histologic data (e.g. K-RAS & B-RAS status), tumor staging and grading (TNM, UICC), therapy documentation, as well as lifetime statistics (overall/progression free survival). Documentation is provided on patient, not lesion level. However, this dataset only covers 78 of the patients of the radiologic dataset.
- **Dataset C** - the combined intersection of dataset A and B. While survival may theoretically be predicted from one timepoint (TP) only (Dataset C), again we expect prior progress to be

predictive, too. Thus, we assume to have at least two timepoints (Dataset C1). When taking into account only patients whose overall survival status (OS) is known, the dataset further reduces (Dataset C2). Experiments for survival prediction were made on this dataset.

As clinical data was only acquired once per patient (Dataset B) the lifetime data in dataset C was calculated relatively to the scan date for patients with multiple radiologic observation timepoints. Table 1 gives an overview on the used datasets.

Table 1: Overview on used raw dataset

| Dataset | Description | $N$ | $N_{patients}$ | Timepoints | Lesions |
|---------|-------------|-----|----------------|------------|---------|
| Dataset A | Radiologic Raw Data | 1235 | 116 | 315 | 458 |
| Dataset A1 | Radiologic Raw Data (TP+FU) | 777 | 94 | 198 | 360 |
| Dataset A2 | Radiologic Raw Data (TP+FU+FU2) | 417 | 55 | 104 | 218 |
| Dataset B | Clinical Raw Data | 135 | 135 | - | - |
| Dataset C | Combined Intersection | 800 | 78 | 211 | 304 |
| Dataset C1 | Combined Intersection (TP+FU) | 496 | 61 | 132 | 231 |
| Dataset C2 | Combined Intersection (TP+FU+OS) | 302 | 33 | 78 | 131 |

## 2.1 Target Variables

We propose the importance of structural image information for two classification targets:

- Tumor Growth - Tumor growth is an important indicator of tumor progression, as well as the overall clinical presentation. This is reflected by the RECIST guideline, which is based on lesion growth assessment, by in turn being the gold standard for solid tumor assessment.

- One-Year-Survival - Survival rates are often based on a large number of previous outcomes for other patients. However, they are influenced by a multitude of factors, e.g. tumor location and genetics, comorbidities and the patient's overall status, which are usually not covered in average survival rates, making it impractical to lock on a concrete estimate. Having an algorithmic assessment which covers the lesion's specifics might be a first step towards a more substantiated estimate.

Labels for tumor growth are assigned according to the RECIST guidelines from Eisenhauer *et al.* (2009), thus assigning a sample $x_i$ with diameter $\varnothing_{i,t}$ at timepoint $t$ a label $y_i$ as follows:

$$y_i = \begin{cases} 1 & \text{if } \varnothing_{i,t+1}/\varnothing_{i,t} \geq 1.2 \\ 0 & \text{otherwise} \end{cases} \tag{1}$$

Opposed to RECIST, this definition uses the current diameter $\varnothing_{i,t}$ as a reference, while RECIST uses the diameter at best response. One-year-survival labels are extracted from the clinical data set relatively to the lesion's scan date. The data distribution can be seen in table 2. Both cases suffer from highly unequal label distributions which has to be considered in algorithm and metrics design.

## 3 Methods

### 3.1 Lesion extraction & preprocessing

First, the dataset was unified by resampling all images to isotropic voxel size ($1.25\,\text{mm}$) using bicubic interpolation. With respect to signal-theory, applying the Lanczos-filter might be better

Table 2: Overview on used raw dataset

| Dataset | N | Positive | Negative |
|---------|---|----------|----------|
| Tumor Growth | 417 | 63 | 354 |
| One-Year-Survival | 302 | 124 | 178 |

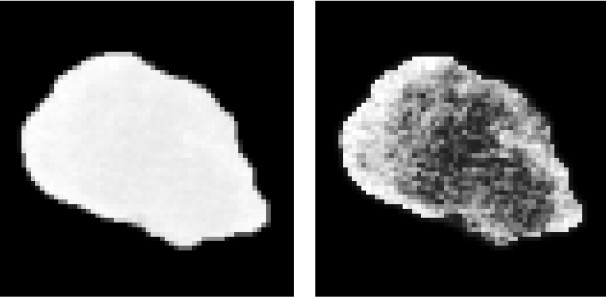

Figure 1: Extracted lesion before (left) and after (right) applying histogram equalization.

substantiated. However, due to the hard segmentation edges we faced disadvantageous harmonics at the tumor borders. The tumor diameter quantiles in our dataset were $\varnothing_{Pr(\varnothing)\leq.1}, \varnothing_{Pr(\varnothing)\geq.9} = 11.3\,\mathrm{mm}, 53.3\,\mathrm{mm}$. Thus, we decided to extract windows of 80 x 80 x 80 mm. As we did not perform a systematic evaluation of voxel and window size with respect to the classification performance, other values might perform better. However, in our first analysis this did not seem to have a major impact on classification performance as long as the lesions are fully contained and sufficiently represented. Segmentation was needed to reduce variability within the dataset with respect to surrounding structures. All lesions did undergo a histogram equalization after extraction. An example lesion is shown in Fig. 1.

## 3.2 Metrics

Within the medical domain high quality demands and precisely known error bars are expected, as errorneously classified samples can have serious consequences for the patient. Thus, choosing the right classifier is a crucial task and we expect this decision to be taken case-by-case. Therefore we provide a variety of measures with different properties (Table 3). The expressiveness of algorithmic metrics depends on a variety of variables. Accuracy, being one of the most often used metric, is only applicable to problems with equal label distributions. As described in sec. 2, we face highly unequal label distribtions, dismissing the use of unbalanced metrics. The F1-/Sørensen-Dice coefficient is widely common, but overweights the positive class, making it prone to changes in the problem definition. Within the clinical setting often sensitivity ($TPR$) and specificity ($TNR$) are preferred. However, these metrics are highly influenced by the classifier's class weight, as an algorithm can be more conservative, hence preferring the negative class, or more optimistic, which in turn privileges the positive class. As a metric with high invariance regarding class imbalances and problem formulation is highly preferable, we propose the use of the $\phi$-/Matthews correlation coefficient, as it a) provides these invariances, b) is robust over a wide range of class-weight choices, and c) is highly common in various scientific domains, as for dichotomous problems it reduces to Pearson's product-moment correlation coefficient $r$.

Table 3: Determined classifier metrics

| metric | abbreviation | formula |
|---|---|---|
| true positive rate, sensitivity, recall | $TPR$ | $\frac{tp}{tp+fn}$ |
| true negative rate, specifity, inverse recall | $TNR$ | $\frac{tn}{tn+fp}$ |
| positive predictive value, precision | $PPV$ | $\frac{tp}{tp+fp}$ |
| negative predictive value, inverse precision | $NPV$ | $\frac{tn}{tn+fn}$ |
| F1 score/Sørensen-Dice coefficient | $F1$ | $\frac{2 \cdot tp}{2 \cdot tp+fp+fn}$ |
| Informedness, Youden's J statistic | $IFD$ | $TPR+TNR-1$ |
| Markedness | $MKD$ | $PPV+NPV-1$ |
| Matthews correlation coefficient/r/$\Phi$-Score | $MCC$ | $\sqrt{IFD \cdot MKD}$ |

### 3.3 Classification with RECIST diameter

Currently there is no established state-of-the-art measure regarding the prediction of treatment outcomes. However, for the radiologic assessment the clinical treatment protocol is based on the Response Evaluation Criteria for Solid Tumors (RECIST). For single lesion assessment, RECIST requires the measurement of the longest lesion diameter within one tumor slice. RECIST in itself is a retrospective measure, only stating wheather a treatment *was responding hitherto*. However, as this assessment is used for therapy adaption based on future expectations, it implies a direction of treatment response, too. It is thus arguable that the RECIST diameter, although being a retrospective measure only, of course *is and can be used predictively*. Thus, for each lesion $x_{i,t}$ with diameter $m_{i,t}$ at timepoint $t$ in **Dataset A** we define a new sample $x'_{i,t}$ as follows:

$$x'_{i,t} = \begin{pmatrix} m_{i,t} \\ m_{i,t-1} \\ m_{i,t} - m_{i,t-1} \\ \frac{m_{i,t}}{m_{i,t-1}} \end{pmatrix} \tag{2}$$

This definition encodes the absolute size and size differences for one timepoint and its successor, as well as the relative growth of the lesion up to the second timepoint. The ground truth is deduced by analyzing one additional successor. As described in sec. 2, the resulting set consists of 417 samples (Dataset A2).

#### 3.3.1 Classifier design

For the final classification a pipeline design is used. The pipeline consists of a z-normalization followed by an ANOVA k-best forward feature selection. Afterwards, the selected features are classified by a random forest (RF). Feature selection and random forest have a variety of hyperparameters, thus classification performance is strongly influenced by their concrete choice. We therefore used 10,000 iterations of 10-fold randomized search cross validation (RSCV) to find the optimal parameter set for all classification pipeline's elements. The classifier metrics are then obtained doing another fit of 10-fold cross-validation with the optimal found parameter set. The metaoptimization includes the number of features $k$ used for RF-classification, the number of estimators $n$, the maximum amount of features used for split at each node, the maximum tree-depth, and the use of balanced vs. unbalanced class weighting.

One could rightly argue, that most parameter sets would perform quite similar in this case, especially as the RECIST-based feature design is quite simple with only 4 dimensions. However, this pipeline design allows for use on much more complex problems. Thus, it can equally be applied to the Radiomics setting without the neccessity of modifications and, in turn, with higher comparability.

### 3.4 Radiomics analysis

For comparibility reasons we also analyze the classification via application of Radiomics feature analysis, as most state-of-the-art approaches utilize Radiomics features for medical image analysis like originally proposed in Kumar *et al.* (2012) and Aerts *et al.* (2014b), or derived variants like Fave *et al.* (2017). We utilize the Radiomics reference implementation from van Griethuysen *et al.* (2017) and further apply the classification pipeline proposed in sec. 3.3.1 on the fully-volumetric segmented lesions from Dataset A2 for tumor growth prediction, respectively Dataset C2 for survival prediction. Again, we introduce difference and ratio features utilizing the definition from eq. 2, overall leading to 4836 features per lesion.

### 3.5 Sparse Characterization through Deep Convolutional Autoencoders

Medical scenarios often suffer from very few data as compared to classical machine vision scenarios datasets are harder to obtain due to data privacy and regulatory issues. Thus, for the application of deep learning a sufficient reduction of data dimensionality is required. Hinton & Salakhutdinov (2006) proposed a sparse encoding based on Restricted Boltzmann Machines which was further adapted and successfully utilized for use with deep convolutional neural networks (DCNNs) (Krizhevsky & Hinton (2011); Krizhevsky *et al.* (2012)). We therefore propose an architecture based on deep convolutional sparse autoencoders for reducing the radiologic lesion image to a minimal sparse vector

representation, which drastically reduces the number of trainable parameters in the classification stage. Additionally, the number of parameters can be limited by classifying single tumor slices instead of fully-volumetric lesions. This, however, assumes that single slices contain enough textural information for classification. While fully-volumetric images likely contain more information, hence volumetric classification presumably would perform better, in our analysis it lead to overfitting and poor test set performance as too few data was available for sufficient training.

### 3.5.1 Network architecture

The training for the final classifier actually employs two architectures, one for the autoencoder pretraining, and one for the final classifer. The autoencoder's general network design requires an input of one lesion at two timepoints. The network itself is divided into two processing columns, where each column represents either the first or the second timepoint. Features are extracted independently from both timepoints and get combined as inputs to the sparse representation layer. After this, the images get split again to form the original column design in reverse order. This two column design performed superior to direct convolution of both images in the first, respectively deconvolution in the last layer with respect to the final performance of the classifier stage. The autoencoder's full architecture is presented in table 4.

After finishing the autoencoder's training, all layers beyond the sparse representation are removed and a simple two-layer classification stage is appended. We analyzed various classifier architectures with alternating numbers of layers and neurons but found one layer with 8 neurons followed by an output layer of 2 neurons with one-hot-encoding to be adequate. This architecture introduces only 218 parameters and, as shown in sec. 4, prove feasible for our scenario to provide sufficient classification and generalization performance.

### 3.5.2 Training methods

Based on the data from sec. 2, overall 417 samples were available for tumor growth prediction (Dataset A2), respectively 302 samples for one-year survival. Thus, the number of free parameters - as mentioned above - is magnitudes higher than the amount of available training samples. We therefore apply Batch Normalization after each convolutional layer, as according to Ioffe & Szegedy (2015) it was shown to be comparably regularizing like dropout and superior in terms of final classification performance. For training we applied Adam-optimization with Nesterov momentum (Nadam) from Dozat (2016), using a batch size of 128 samples. As discussed in sec. 2.1, the class distribution in our datasets is highly imbalanced. To tackle this problem, the training process is stratified by applying class importance sampling. Therefore each sample $x_i \in X = \{x_1, ..., x_m\}$ was assigned a sampling probability $p(x_i)$ with:

$$p(x_i) = \frac{\frac{1}{Pr(y=y_i)}}{\sum_{k=0}^{n} \frac{1}{Pr(y=y_k)}} \tag{3}$$

Additionally, we applied a modified version of the exact important sampling algorithm from Katharopoulos & Fleuret (2017), transforming the exact importance into a logarithmic equal distribution across all training samples at the end of each epoch, as this modification showed higher stability in the training process. For the final results we used 4 splits of grouped cross validation with the patient being the grouping parameter.

### 3.5.3 Reasoning

A major difficulty when using deep learning for medical image classification arises from the non-interpretability. Neural networks alone are intangible, intransparent and incomprehensible. This applies particularly for end users without deeper technical knowledge, but nevertheless stays true for computer scientists with a well-substantiated knowledge base in machine learning and neural networks, as neural networks apply an intuitively nearly unmanageable amount of nonlinear functions and embeddings. Therefore, it is neccessary (and demanded) to visualize, which factors are responsible for the network's prediction. We propose a visualization of indicative regions based on the algorithm from Simonyan *et al.* (2013). The visualization is created with guided backpropagation of the network activation from the output layer back to the network's input. This allows to visualize regions which are indicative for specific labels, e.g. tumor growth. An example visualization can be found in Fig. 2.

Table 4: Autoencoder neural network design

|    | type | filters | stride | regularization | output | # parameters |
|----|------|---------|--------|----------------|--------|--------------|
|    | input | – | – | BN | 2x64x64x1 | 0 |
|    | lambda | – | – | BN | 0: 64x64x1
1: 64x64x1 | 0 |
| 2x | conv | 5x5 | 1x1 | BN | 64x64x32 | 960 |
|    | pool | 2x2 | 2x2 | – | 32x32x32 | – |
|    | conv | 5x5 | 1x1 | BN | 32x32x48 | 38640 |
|    | pool | 2x2 | 2x2 | – | 16x16x48 | – |
|    | conv | 3x3 | 1x1 | BN | 16x16x64 | 27968 |
|    | pool | 2x2 | 2x2 | – | 8x8x64 | – |
|    | conv | 3x3 | 1x1 | BN | 8x8x96 | 55776 |
|    | pool | 2x2 | 2x2 | – | 4x4x96 | – |
|    | conv | 3x3 | 1x1 | BN | 4x4x128 | 111232 |
|    | reshape | – | – | – | 4096 | – |
|    | dense | – | – | BN | 20 | 82020 |
| 2x | dense | – | – | BN | 2048 | 51200 |
|    | reshape | – | – | – | 4x4x128 | – |
|    | deconv | 3x3 | 1x1 | BN | 4x4x96 | 111072 |
|    | up | 2x2 | 2x2 | – | 8x8x96 | – |
|    | deconv | 3x3 | 1x1 | BN | 8x8x64 | 55616 |
|    | up | 2x2 | 2x2 | – | 16x16x64 | – |
|    | deconv | 3x3 | 1x1 | BN | 16x16x48 | 27888 |
|    | up | 2x2 | 2x2 | – | 32x32x48 | – |
|    | deconv | 5x5 | 1x1 | BN | 32x32x32 | 38560 |
|    | up | 2x2 | 2x2 | – | 64x64x32 | – |
|    | deconv | 5x5 | 1x1 | BN | 64x64x1 | 932 |
|    | out | – | – | – | 2x64x64x1 | – |

Table 5: Classifier neural network design

|    | type | filters | stride | regularization | output | # parameters |
|----|------|---------|--------|----------------|--------|--------------|
|    | input | – | – | BN | 2x64x64x1 | 0 |
|    | lambda | – | – | BN | 0: 64x64x1
1: 64x64x1 | 0 |
| 2x | conv | 5x5 | 1x1 | BN | 64x64x32 | 960 |
|    | pool | 2x2 | 2x2 | – | 32x32x32 | – |
|    | conv | 5x5 | 1x1 | BN | 32x32x48 | 38640 |
|    | pool | 2x2 | 2x2 | – | 16x16x48 | – |
|    | conv | 3x3 | 1x1 | BN | 16x16x64 | 27968 |
|    | pool | 2x2 | 2x2 | – | 8x8x64 | – |
|    | conv | 3x3 | 1x1 | BN | 8x8x96 | 55776 |
|    | pool | 2x2 | 2x2 | – | 4x4x96 | – |
|    | conv | 3x3 | 1x1 | BN | 4x4x128 | 111232 |
|    | reshape | – | – | – | 4096 | – |
|    | dense | – | – | BN | 20 | 82020 |
|    | dense | – | – | BN | 8 | 200 |
|    | softmax | – | – | – | 2 | 18 |

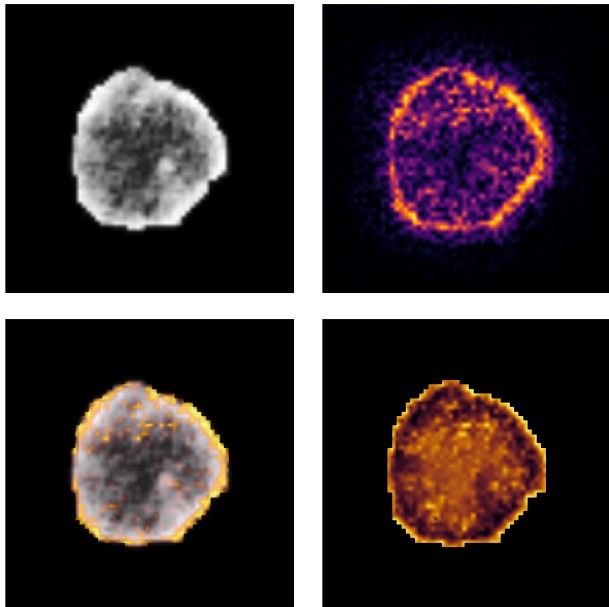

Figure 2: Saliency map based reasoning of classifier decisions with a modified version of the algorithm from Simonyan *et al.* (2013). Left: Lesion image (top) and overlay visualization of input regions predictive for future tumor growth (bottom). Expectedly, lesion marginalization and contrast enhancement are important predictors for future progress. However, inner structure is marked as predictive, too. Right: Raw saliency map (top), highlighting the importance of inner structure, with only the most inner necrosis being non-predictive for growth. Bottom: Intensity influence adjusted saliency map, again highlighting the predictive inner structure.

## 4 Results

We systematically analyzed the predictive performance of all mentioned classifiers with respect to the metrics discussed in sec. 3.2. The results for tumor growth prediction are shown in table 6, results for one-year-survival prediction can be found in table 7. Significant improvements with respect to all reference classifiers ($p < 0.05$) are marked with a star (*). The confidence intervals were computed using 10,000 iterations of bootstrapping as proposed by Efron (1992). Significance was tested by employing a two-tailed Z-test.

| Metric | DL | | RECIST | | Radiomics | | Informed Guess | | sig. |
|---|---|---|---|---|---|---|---|---|---|
| TPR | **.743** | *[.648,.831]* | .430 | [.302,.552] | .363 | [.245,.483] | .154 | [ .082,.237] | * |
| TNR | .768 | *[.730,.806]* | .864 | [.827,.901] | **.912** | [.881,.940] | .847 | [ .815,.879] | |
| PPV | .366 | *[.292,.439]* | .359 | [.250,.468] | **.425** | [.292,.560] | .153 | [ .077,.235] | |
| NPV | **.944** | *[.918,.965]* | .894 | [.861,.926] | .890 | [.858,.922] | .847 | [ .813,.880] | * |
| F1 | **.490** | *[.412,.561]* | .390 | [.280,.490] | .390 | [.268,.505] | .152 | [ .081,.226] | * |
| IFD | **.511** | *[.405,.606]* | .296 | [.167,.425] | .277 | [.153,.404] | .000 | [-.083,.080] | * |
| MKD | .311 | *[.239,.390]* | .258 | [.148,.383] | **.318** | [.183,.452] | -.002 | [-.083,.092] | |
| MCC | **.400** | *[.314,.480]* | .273 | [.159,.400] | .294 | [.166,.420] | .000 | [-.075,.082] | * |
| AUC | **.784** | *[.735,.833]* | .744 | [.674,.810] | .737 | [.669,.803] | .500 | [ .461,.545] | |

Table 6: Results on tumor growth prediction using our proposed deep learning approach, RECIST diameter, Radiomics or informed guessing ($H_0$).

Regarding tumor growth prediction, our approach performed significantly better than any other tested classifier in terms of $TPR$, $NPV$, $F1$, $IFD$ and $MCC$. Regarding $AUC$, it also provided the best results (.784, $CI_{95} = [.735, .833]$), though these results were not significantly better than

RECIST-based or Radiomics prediction (.744/.737). All classifiers were significantly better than $H_0$ for all tested metrics, except $TNR$, which follows from the unequal label distribution (see table 2). Expectedly, the concrete class weight choice had a major impact on $TPR$, $TNR$, $PPV$, $NPV$ and, while being less severe, $F1$-score for all tested classifiers.

Regarding one-year-survival prediction, again all tested classifiers were significantly better than $H_0$ for all tested metrics, except $TNR$. Our proposed deep learning approach, however, provided significant superiority to any other tested classifier with respect to $TNR$, $PPV$, $IFD$, $MKD$ and $MCC$. While the results for $IFD$ and $AUC$ were the best among all tested classifiers, the differences were not significant better than RECIST-based prediction ($IFD_{DL} = .387$, $CI_{95} = [.290, .484]$ vs. .332 for RECIST diameter; $AUC_{DL} = .710$, $CI_{95} = [.645, .773]$ vs. .688 for RECIST diameter). Radiomics classification was highly inferior to DL- and RECIST-based prediction. However, as the dataset contained lesions from only 33 patients and group split was done along this axis, the results for random forest classification with Radiomics are likely not representative for larger datasets and partly caused by our classification pipeline definition including a feature selection step, leading to intense overfitting on the training subset. This is especially likely as $m \gg n$ holds true, with $m$ being the number of features and $n$ being the number of samples. Also the modification of split numbers or split sizes did not lead to significant improvements, while the reduction on 2D-shape features expectedly lead to results comparable with RECIST-based classification. The receiver operating characteristic for growth and survival prediction with our deep learning approach are shown in Fig. 3.

| Metric | DL | | RECIST | | Radiomics | | Informed Guess | | sig. |
|---|---|---|---|---|---|---|---|---|---|
| TPR | .462 | [.368,.547] | **.717** | [.593,.638] | .566 | [.482,.648] | .411 | [.328,.496] | |
| TNR | **.927** | [.882,.963] | .612 | [.538,.683] | .620 | [.550,.689] | .589 | [.514,.661] | * |
| PPV | **.815** | [.721,.902] | .561 | [.482,.646] | .507 | [.425,.592] | .412 | [.331,.496] | * |
| NPV | .712 | [.655,.768] | **.756** | [.684,.830] | .670 | [.599,.739] | .590 | [.511,.668] | |
| F1 | .586 | [.497,.667] | **.630** | [.557,.695] | .534 | [.459,.608] | .410 | [.336,.487] | |
| IFD | **.387** | [.290,.484] | .332 | [.225,.434] | .182 | [.072,.288] | .000 | [-.114,.113] | |
| MKD | **.528** | [.419,.634] | .321 | [.218,.426] | .178 | [.069,.277] | .000 | [-.108,.109] | * |
| MCC | **.449** | [.344,.541] | .321 | [.219,.423] | .180 | [.063,.288] | .000 | [-.102,.102] | * |
| AUC | **.710** | [.645,.773] | .688 | [.629,.740] | .568 | [.504,.628] | .500 | [.440,.557] | |

Table 7: Results for one-year-survival prediction with our deep learning approach, RECIST diameter, Radiomics and informed guess ($H_0$).

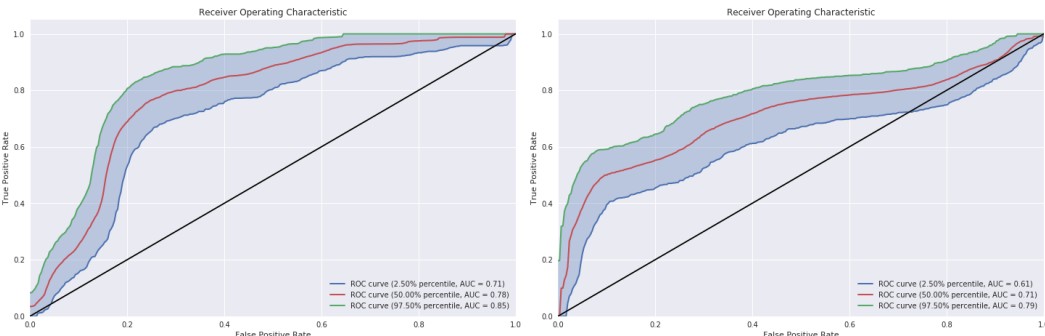

Figure 3: Receiver Operating Characteristic for tumor growth (left) and survival prediction (right) with our proposed deep learning approach.

## 5 Discussion

Radiomics was shown to be beneficial for tasks like lung or head and neck cancer assessment, prediction of lymph node metastases or breast cancer risk Aerts *et al.* (2014a); Huang *et al.* (2016); Li *et al.* (2016). However, in our experiments Radiomics did not provide significant benefits over RECIST-based prediction for liver lesion assessment. More sophisticated Radiomics approaches

might perform better on the given problem. Additionally, our data contained various scanners and their respective configurations, e.g. varying kernels, image resolution or slice thicknesses, as well as some differences in contrast agent phase. Considering the data heterogeneity, the used dataset may be too small for the given problem. It was also shown that many Radiomics features are highly influenced by different scaner configurations (e.g. Mackin *et al.* (2015); Leijenaar *et al.* (2013)). We expect the Radiomics results to be significantly better assuming the use of unified scan protocols and/or feature correction.

The results for our proposed deep learning approach imply that the radiologic tumor phenotype itself encodes information beneficial for predicting tumor progress, as well as the patient's final outcome. While predicting the lesion's growth seems possible from structural information, growth itself still is only an intermediate result, as it is often, but not neccessarily, linked to the patient's final outcome. It is rather of clinical value to understand *which* structural specifics actually are predictive for tumor growth or the final outcome, and how to practically acquire and interpret these values within the clinical workflow. The visualization proposed in sec. 2 may be a first step in this direction.

Our results give at least some indication that prediction of overall survival may be possible and feasible using deep neural networks, which may have high clinical value. Similar approaches for other clinical domains include the work from Yao *et al.* (2016, 2017) and Nie *et al.* (2016). While the accuracy of these approaches is still far away from a to-the-day estimate, the prediction of the patient's survival, whether for clinical application or other domains, could have substantial ethical implications which have to be discussed publicly, as the media attention on the paper from Avati *et al.* (2017) has shown.

One major drawback of the used datasets is data heterogeneity, which applies to the radiologic as well as the clinical presentation. The lesions in our dataset underwent a wide range of therapies, including 5FU, FOLFOX, XELOX and surgical intervention. As it is very likely that different therapies show different growth patterns and/or textural structures, training the classifier on a more homogenous dataset could presumably provide better performance. Additionally, like many clinical datasets, our dataset is considerably small for a deep learning task. Hence, at least some architecture rethinking and regularization is required, again impeding the actual training problem. Large, standardized datasets like Armato *et al.* (2011) or Aerts *et al.* (2015) might help to face these problems, and thus, have the potiential of leading to an overall improved patient healthcare.

## 6 Conclusion

Our study shows that decision support for oncologic assessment is continously developing, benefitting from the integration of new methods and technologies. (Semi)-Automatic treatment assessment remains a highly interesting research field, promising further improvements within the next years. The automatic radiologic assessment with deep neural networks might give additional, valuable information based on textural information, which in turn might enhance patient treatment and thus overall healthcare. With the rise of new methods, ultimately, neural networks might even be able to give a new insight into biological processes.

**Acknowledgments**

This work has received funding from the German Federal Ministry of Education and Research as part of the PANTHER project under grant agreement no. 13GW0163A.

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
