# OpenReview forum: "Predicting Lesion Growth and Patient Survival in Colorectal Cancer Patients using Deep Neural Networks"
_MIDL.amsterdam/2018/Conference — MIDL 2018 Poster_

### Review · AnonReviewer3 · 2018-05-06
**Little methodological novelty with decent validation**

**Rating:** 2
**Confidence:** 2

**Review:**

This paper proposes a deep neural networks-based classification method to predict lesion growth and patient survival with CT images of colorectal cancer patients. The method was well validated in the experiments. The overall writing is satisfactory with few typos. The evaluation metrics are extensive. However, technical contribution and the demonstration of saliency maps are some kind of weak.

Pros:
1, the descriptions of the data are detailed
2, three traditional methods were compared with the proposed framework
3, the quantitative evaluations are comprehensive with many different metrics

Cons:
1, The motivation of using an auto-encoder followed by a classifier is not clarified. Why not using a network for classification directly?
2, Several typos are found in the manuscript. E.g., “provide ,,a standard set”, “Therefore….”, etc.
3, The saliency map was demonstrated with only one example. It’s better to show results with different classes of growth/non-growth and survival/non-survival prediction.


**Special Issue:**

No

---

> ### Comment · ~Alexander_Katzmann1 · 2018-06-11
> **Why used an auto-encoder design?**
>
> Dear reviewer,
>
> Thank you very much for your review. I could easily follow your points and agree with you.
>
> Regarding your question on the motivation of using an auto-encoder: It arises from the fact that we had only few, partially highly correlated training data. As - opposed to segmentation tasks or similar - our data is not easily decomposable, we faced the problem of having masses of free parameters, which we could only set against very few training data samples in relation. This in turn led to overfitting, which we tried to prevent by pretraining an autoencoder with very few neurons, thus forming a bottleneck within the network and enforcing a very general dimensionality reduction.
>
> You are right, that this should have been highlighted some more within the publication. I hope it answers your question adequately.
>
> Again thank you very much for your review. I will mind your issues for future publications. In case you might have any further questions, please feel free to ask at any time.
>
> Best regards,
> Alexander.

---

### Review · AnonReviewer2 · 2018-05-09
**This work presents deep learning-based network for tumor growth prediction and one-year survival prediction for CRC metastatic liver tumors. Different approaches are tested and compared showing the benefit of the system.**

**Rating:** 3
**Confidence:** 2

**Review:**

The paper is well-written (with some parts of the text overextended). Information is provided to fully understand the concept and the method being tested. Would be helpful to include a sketch of the system (2 time point entry). The presented method is general and can be applied to different applications making it interesting for the conference community.

Pros:
-	The presented system shows superior results over other methods.
-	Statistical measurements and significance were carefully computed
-	High quality evaluation with unique dataset.

Cons:
-	Some parts of the text can be reduced or written shortly (e.g. metrics, discussion).
-	The description of the datasets used and the entire procedure for handling the data is overly extended.
-	The novelty of the method is weak


Minor comments:
Section 3.1 – the quantiles equation seems to have a wrong inequality direction for the 0.9 (?).
It is not stated what the manual segmentation was needed for – would a simple bounding box work?
Equation 2 – should state "m" in the following text.
Figure 3 – text is too small.
Check reference format.



**Special Issue:**

No

---

> ### Comment · ~Alexander_Katzmann1 · 2018-06-11
> **Thanks + answer to your question**
>
> Dear reviewer,
>
> Thank you very much for your detailed reply. You are addressing some important issues here, which I payed attention to within in the current revision. As I am still a young member of the research community, your feedback helps me very much to improve my work. I will take care of the issues within my future papers.
>
> Regarding your question whether a simple bounding box would work: Probably not this good. We first tried to train our network without any segmentations, resulting in a higher amount of overfitting. Alternatively, more regularization was needed, thus reducing the classification performance. All together variance reduction was in our case the key to successful classification.
>
> I hope this answers your question. In the next publication I will probably highlight this finding some more. In case of any further questions, feel free to contact me at any time.
>
> Again many thanks for your review.
>
> Best regards,
> Alexander.

---

### Review · AnonReviewer1 · 2018-05-17

**Rating:** 3
**Confidence:** 2

**Review:**



**Special Issue:**

No

---

### Decision · Program_Chairs · 2018-05-15
**Paper22 Acceptance Decision**

Poster